# Essential Oil Constituents as Anti-Inflammatory and Neuroprotective Agents: An Insight through Microglia Modulation

**DOI:** 10.3390/ijms25105168

**Published:** 2024-05-09

**Authors:** Nikola M. Stojanović, Pavle J. Ranđelović, Maja Simonović, Milica Radić, Stefan Todorović, Myles Corrigan, Andrew Harkin, Fabio Boylan

**Affiliations:** 1Department of Physiology, Faculty of Medicine, University of Niš, 18000 Niš, Serbia; nikola.st90@yahoo.com (N.M.S.); pavleus@gmail.com (P.J.R.); 2Department of Psychiatry, Faculty of Medicine, University of Niš, 18000 Niš, Serbia; simonovicmaja@gmail.com; 3University Clinical Centre Niš, 18000 Niš, Serbia; milica91nis@ymail.com (M.R.); todorovicstefan815@gmail.com (S.T.); 4Department of Oncology, Faculty of Medicine, University of Niš, 18000 Niš, Serbia; 5School of Pharmacy and Pharmaceutical Sciences, Trinity College Dublin, Dublin 2, Ireland; corrigmy@tcd.ie (M.C.); aharkin@tcd.ie (A.H.); 6Trinity Biomedical Sciences Institute (TBSI) and The Trinity Centre for Natural Product Research (NatPro), D02 R590 Dublin, Ireland

**Keywords:** essential oils, microglia, neuroinflammation, neurodegenerative disease

## Abstract

Microglia are key players in the brain’s innate immune response, contributing to homeostatic and reparative functions but also to inflammatory and underlying mechanisms of neurodegeneration. Targeting microglia and modulating their function may have therapeutic potential for mitigating neuroinflammation and neurodegeneration. The anti-inflammatory properties of essential oils suggest that some of their components may be useful in regulating microglial function and microglial-associated neuroinflammation. This study, starting from the ethnopharmacological premises of the therapeutic benefits of aromatic plants, assessed the evidence for the essential oil modulation of microglia, investigating their potential pharmacological mechanisms. Current knowledge of the phytoconstituents, safety of essential oil components, and anti-inflammatory and potential neuroprotective effects were reviewed. This review encompasses essential oils of *Thymus* spp., *Artemisia* spp., *Ziziphora clinopodioides*, *Valeriana jatamansi*, *Acorus* spp., and others as well as some of their components including 1,8-cineole, *β*-caryophyllene, *β*-patchoulene, carvacrol, *β*-ionone, eugenol, geraniol, menthol, linalool, thymol, *α*-asarone, and *α*-thujone. Essential oils that target PPAR/PI3K-Akt/MAPK signalling pathways could supplement other approaches to modulate microglial-associated inflammation to treat neurodegenerative diseases, particularly in cases where reactive microglia play a part in the pathophysiological mechanisms underlying neurodegeneration.

## 1. Introduction

Microglia, resident macrophage-like cells in the brain, play a pivotal role in the innate immune response, contributing to homeostatic and reparative functions [1]. These cells can become activated in response to injury or exposure to various immunogenic stimuli such as pathogens or abnormal protein aggregates, triggering the release of proinflammatory cytokines, chemokines, and reactive oxygen species (ROS) [2]. Microglia function as antigen-presenting cells and express recognition receptors including CD14, TLR-2, and TLR-4 (Toll-like receptor-2, -4), initiating an immune-like response via signalling pathways (see Figure 1). Microglia can be categorised into two distinct types: classical (M1) or alternative (M2). M1 microglia release inflammatory mediators, leading to inflammation and neurotoxicity, while M2 microglia release anti-inflammatory mediators, engage in phagocytosis of cell debris and misfolded proteins, promote extracellular matrix reconstruction and tissue repair, and support neuron survival through the production of neurotrophic factors [3] (see Figure 2). Notably, a spectrum of intermediate phenotypes exists between M1 and M2, and microglia can transition from one phenotype to another. Consequently, microglia are considered as a double-edged sword, capable of exerting both detrimental and beneficial effects in the brain [2]. Maintaining a balanced polarisation between M1 and M2 states is crucial for the progression of chronic neurodegenerative diseases [2]. Chronic and excessive microglial activation results in the production of excessive proinflammatory molecules such as cytokines and ROS, contributing to neuronal damage and cell death [4]. The release of inflammatory mediators can disrupt normal neuronal functioning, impair synaptic plasticity, and contribute to the progression of neurodegeneration, ultimately leading to functional deficits such as cognitive and motor impairments [5].

Considering that CNS inflammation and degeneration are an enormous issue in medicine nowadays, the current review aimed to evaluate and summarise the findings of essential oils (EOs) in different experimental models. This was conducted through a detailed analysis of the publications collected after the keywords “essential oils” and “microglia” were entered into the databases. A literature survey using the Scopus, PubMed, and SciFinder search browsers (from 1990 to April 2024) resulted in the retrieval of around 65 papers addressing this topic. Papers found by this search were excluded because specific keywords were mentioned but deemed outside the scope of this this review. Other exclusion criteria included inadequate sources (review papers, conference abstracts, and patents).

## 2. Signalling Pathways Associated with Microglia

Microglia play a critical role in removing toxic protein aggregates such as amyloid-beta (A*β*) in Alzheimer’s disease (AD) and *α*-synuclein in Parkinson’s disease (PD) through phagocytosis and degradation. This clearance reduces the accumulation of pathological proteins, potentially slowing disease progression. Microglia also release neurotrophic factors and anti-inflammatory molecules that support neuronal survival and repair [6]. In AD, despite possessing A*β* clearance receptors, ineffective microglial-associated clearing mechanisms and proinflammatory processes contribute to disease advancement [3]. Genome-wide association studies have identified AD risk genes primarily expressed in microglia such as TREM2, CD33, BIN1, and CR1 [7]. AD patients exhibit impairments in microglial function including reduced phagocytic activity coinciding with A*β* plaque deposition [8]. In PD, extracellular *α*-synuclein can activate TLR2 in microglia, triggering inflammatory responses. Targeting *α*-synuclein and TLR2 may offer therapeutic potential for modifying neuroinflammation in PD and other conditions like Niemann–Pick disease type C1 (NPC1), multiple sclerosis, AD, and stroke [9,10,11].

Microglial inflammatory responses involve diverse signalling pathways including TLR/MyD88/NF-κB, Janus kinase/signal transducer and activator of transcription (JAK/STAT), and p38 mitogen-activated protein kinase (MAPK), which are implicated in cell proliferation, differentiation, apoptosis, and immune regulation [12]. The activation of the TLR4/MyD88/NF-κB signalling pathway in microglia contributes to the upregulation of proinflammatory factors, exacerbating injury within the central nervous system (CNS) [13]. Signalling mediated through the JAK/STAT pathway is initiated by various cytokines, leading to the activation of a proinflammatory phenotype [14]. Targeting p38*α* MAPK complements other pathways in regulating microglia and microglial-associated inflammation [15]. This pathway is identified as a crucial regulator of IL-1*β* and TNF*α* production by microglia in response to TLR ligands, extending to activators such as A*β* 1-42. Disruption of the blood–brain barrier (BBB) can increase the infiltration of peripheral monocytes into the brain and stimulate microglial activity [16]. The P2X7 receptor, responding to extracellular ATP, presents as a therapeutic target with potential for regulating neuroinflammation [17]. Targeting potassium channels, particularly Kv1.3, is crucial in modulating microglial activation and neuroinflammation [18]. Additionally, sex hormones including oestradiol influence the proinflammatory responses of microglia [19]. Modulating microglia offers diverse approaches to regulating neuroinflammation, holding promise for the development of new treatments for various neurological diseases with a neuroinflammatory component. 

## 3. The Potential of Essential Oils to Modulate Microglia and Microglial-Associated Inflammation

Essential oils (EOs) represent distinctive mixtures of components that can be extracted from aromatic plants comprising a variety of constituents with different chemical structures including mono-, sesqui-, and di-terpenes, sulphur-containing molecules and phenylpropanoids presenting alcohols, simple phenolics, esters, and ketones, among others in their structure [20]. The number of studies dealing with the activity of EOs and their constituents in the CNS is increasing [21]. Relatively modest research has been conducted on the impact of EOs and their constituents on microglia functioning. On the other hand, EOs should be focused on as potential neuroprotective plant extracts since, due to their lipophilicity, they can penetrate a BBB and exert their action. Furthermore, the complexity of a mixture such as EO allows for a multitarget approach, meaning that different compounds (classes of compounds) could exert their beneficial action through different target molecules/pathways. This approach can benefit from both the synergistic and additive effects of compounds present within a mixture, thus exerting a better activity. Plants used as a source of EOs for the investigation of their influence on microglia were species belonging to the families of Apiaceae [22], Asteraceae [23,24,25], Cupressaceae [26], Ericaceae [27], Lamiaceae [28], Valerianaceae [29], and others [30]. Some plant species are used traditionally, especially in Chinese medicine, for treating CNS disorders [31], while some have previously been proven to possess anti-inflammatory activity [30,32,33]. Additionally, plants investigated for their impact on microglia are occasionally used daily in cooking and/or for treating some non-CNS-related disorders [34].

Essential oils (EOs) and their constituents are recognised for their potential cognitive-enhancing effects, which involve attenuating the neurotoxicity of A*β*, reducing oxidative stress by scavenging reactive oxygen species (ROS), modulating the central cholinergic system, and regulating microglia-mediated neuroinflammation [35]. Demonstrating anti-inflammatory properties, EOs can reduce microglial activation and proinflammatory responses, thereby mitigating the detrimental effects of inflammation on neuronal cells [36]. Essential oils have been found to inhibit the activation of NF-κB and MAPKs, leading to a decrease in the production of proinflammatory cytokines and chemokines in microglia [3]. Furthermore, EOs modulate the expression of inflammatory mediators including inducible nitric oxide synthase (iNOS) and cyclooxygenase-2 (COX-2), which contribute to the production of gaseous and eicosanoid proinflammatory molecules, respectively [36,37].

EOs may also impact neuroinflammation through the gut–brain axis [38]. Studies suggest that EOs modulate intestinal permeability and gut microbiota, potentially indirectly influencing microglia-related function by improving gut health and reducing gut-derived inflammation. Beyond their anti-inflammatory effects, essential oils exhibit neuroprotective properties, shielding neuronal cells from damage induced by ischemia/reperfusion-induced brain injury. These neuroprotective effects may be attributed, at least in part, to the ability of EOs to inhibit the release of neurotoxic factors from microglial cells [36].

## 4. Key Essential Oils of Interest and Their Anti-Inflammatory Activity

Detailed analysis has been conducted on several plant species belonging to genera such as *Artemisia*, *Polygonum*, *Thymus*, *Zingiber*, and others in order to evaluate the anti-inflammatory, antioxidative, and neuroprotective properties of their constituent EOs and individual components. Essential oils and components from *Artemisia dracunculus*, *Artemisia herba-alba*, *Acorus gramineus*, *Patrinia scabiosaefolia*, *Pogostemon cablin*, *Schisandra chinensis*, *Thuja orientalis*, *Thymus vulgaris*, *Thymus zygis* L. subsp. *sylvestris*, and *Ziziphora clinopodioides* are reputed to contain components that possess anti-inflammatory and neuroprotective properties. As EOs are, as above-mentioned, a mixture of many constituents, it is difficult to attribute the observed activity to a single constituent. Potential synergism or additive effects of constituents are likely. Some other constituents are less frequently encountered, or their properties still need to be elucidated; thus, in the present review, the results of studies evaluating the activity of EOs and their pure constituents are summarised. Components were selected considering their intersection with microglial regulation and inflammation and to assess putative mechanisms of action.

### 4.1. Lamiaceae Family Associated EOs

The Lamiaceae family is widely recognised as a rich source of EOs with various medicinal properties [39]. These properties encompass a diverse range including antioxidant, anti-inflammatory, immunoregulatory, antiallergic, neuroprotective, antidepressant, cholinergic, sedative, analgesic, antipyretic, antitussive, anti-asthmatic, decongestant, antidiabetic, antihypertensive, antiseptic, antiparasitic, anthelmintic, anti-angiogenic, anti-hepatotoxic, antiviral, antimicrobial, antiemetic, antifungal, insecticidal, antitumour, and anticancer effects. Chemical composition, which can vary even within the same species, contributes to the diverse activities exhibited by these oils [40].

The *Thymus* genus, a part of the Lamiaceae family, includes over 300 perennial herbaceous plants known for their aromatic qualities and medicinal properties [41]. Thyme oil, extracted from Thymus plants, is extensively used in the food, pharmaceutical, and cosmetic industries, and it is reported to possess antioxidative, anti-inflammatory, antitumour, and antimicrobial properties [42]. The chemical composition of thyme EOs varies, with thymol, carvacrol, and linalool consistently identified as primary components in *Thymus vulgaris* [43]. These compounds have demonstrated medicinal properties including potential benefits for treating brain degenerative diseases [44]. The chemotypes of *T. vulgaris*, linalool (69.2% of EO), geraniol (54.9% of EO), and thujanol (39.4% of EO) are considered promising candidates for treating inflammation and related neurodegeneration, with their ratio being a crucial factor [37]. Linalool, geraniol, and thujanol have been reported to decrease the inflammatory response of BV-2 microglia to LPS, affecting TNF-*α* and IL-6 secretion via the NF-κB signalling pathway [37].

*Thymus zygis* L. subsp. *sylvestris*, a plant species found in central Portugal, is traditionally used for culinary purposes, as a preservative, and for medicinal applications such as treating colds, sore throats, and promoting wound healing [45]. Recent studies have evaluated the effects of *T*. *zygis* subsp. *sylvestris* EO and its main components, p-cymene, thymol, and carvacrol, on BV-2 microglial viability (Table 1) [28]. The study also assessed the action of the EO and these compounds on LPS-induced NO production in RAW 264.7 macrophages and BV-2 microglial cells, reporting that both the EO and its constituents had more potent inhibitory effects on NO production in microglial cells than in macrophages. The combination of constituents exhibited more pronounced effects than each individual compound, with thymol identified as the most potent constituent. However, thymol and carvacrol showed higher cytotoxicity compared to the EO, suggesting that their inhibitory effect on NO production might be associated with a loss of cell viability [28].

Among the members of the Lamiaceae family, the Ziziphora species serves as a prototypical representative, with over 30 species found across Asia, Europe, and Africa, which play crucial roles in pharmaceutical, chemical, traditional, and folk medicines. The phytochemical composition of Ziziphora includes monoterpene EO, triterpenes, and phenolic substances. *Ziziphora clinopodioides* is a plant species rich in EO, found to contain pulegone (42.1%), isomenthone (9.7%), 1,8-cineole (8.22%), piperitone (7.35%), and neomenthole (5.9%) [46]. Different extracts obtained from *Z. clinopodioides* have been reported to possess neuroprotective properties in a rat model of AD. Other in vitro experiments have reported antioxidant properties [46]. The antioxidant activity is attributed to the significant presence of 1,8-cineol [47]. The anti-inflammatory properties of *Z. clinopodioides* EO have been assessed in vitro in BV-2 wild-type and acyl-CoA oxidase 1 (ACOX1)-deficient (Acox1−/−) microglial cells (Table 1) [24]. During the first 24 h of exposure, cytotoxicity in both cell lines was observed when cells were exposed to 50 μL/mL of the EO. With prolonged cultivation of up to 72 h in the presence of 0.5 μL/mL of EO, the cytotoxicity was found to be more pronounced in the ACOX1 deficient cell line [47]. These results suggest that the EO has a more damaging influence on ACOX1 deficient cells that carry a specific microglial peroxisomal *β*-oxidation defect, thus affecting the ability of these cells to regulate oxidative stress and inflammation [48]. In subsequent experiments, *Z. clinopodioides* EO decreased the catalase (CAT) activity in healthy cells while having no impact on enzyme activity in the ACOX1 deficient cells, suggesting that the change in cell viability in the model does not involve this enzyme. On the other hand, EO reduced the accumulation of long-chain fatty acids and the generation of proinflammatory cytokines, which are a characteristic change observed in the model [46]. In contrast to *T. zygis* subsp. *sylvestris* EO, the presence of thymol is scarce in *Z. clinopodioides* EO [46].

*Pogostemon cablin* (Patchouli, Lamiaceae) is a perennial flowering plant that grows in warm tropical climates [34]. The EO obtained from patchouli is mainly used in the perfume industry and aromatherapy. The most important EO components include germacrene-B, patchoulol, and norpatchoulenol [34]. Apart from its use in industry, it is a part of several Chinese herbal remedies for treating gastroenteritis, arthritis, fever, cold, and headache [34]. The exposure of BV-2 cells to the EO obtained from patchouli produced cytotoxicity at concentrations higher than 145 μg/mL [34]. These results are preliminary, and further studies are required to determine its potential to modulate microglial function.

### 4.2. Asteraceae Family Associated EOs

Plants belonging to the *Artemisia* genus are known to be rich in EOs [49], and several EOs from species belonging to this genus have been investigated for their influence on the microglial inflammatory response. These plants have been used for centuries by different nations, and their medicinal properties are reported to be associated with the EO content [24,50]. Reports have indicated that *A. dracunculus* L. possesses medicinal properties including antibacterial, antifungal, anti-inflammatory, antioxidant, and antitumour activity [50]. The EO of *A. dracunculus* consists of several main components that vary in their constituent abundance and can include estragole, also known as methyl chavicol or p-allyl anisole (concentrations of 40–85%), sabinene (up to 35%), methyl eugenol (up to 25%), and elemicin (up to 57%). Other compounds present in the oil at concentrations exceeding 10% include *β*-ocimene, cis-ocimene, *α*-trans-ocimene, limonene, trans-anethole, *α*-phellandrene, and capillene. This EO was found to enhance the antioxidant capacity of BV-2 microglia by stimulating CAT and superoxide dismutase (SOD) activity (Table 1) in the cell culture over 24 and 48 h periods of exposure [24]. The EO in question was found to be similar to the ones grown in high altitude in Armenia, and was rich in estragole, whose concentration reached approximately 85% [24].

*Artemisia herba-alba* is widely used in traditional medicine to treat infection and inflammatory-based disorders such as colds, coughing, bronchitis, and diarrhoea [23]. Although many publications relate to the characterisation of *A. herba*-alba EOs, their safety has been rarely investigated [23]. The EO is comprised mainly of oxygen-containing monoterpenes, which include 1,8-cineole (20.1%), *β*-thujone (25.1%), *α*-thujone (22.9%), and camphor (10.5%) [23]. Exposure of BV-2 microglial cells to *A. herba-alba* EO produces a concentration dependent effect on viability, where the lowest tested concentration of 0.32 μL/mL was the only non-cytotoxic one. On the other hand, all tested concentrations from 1.25 to 0.16 μL/mL reduced BV-2 cell NO production in response to stimulation with LPS (Table 1). These results suggest that the activity might not always arise from the direct action of the EO, but rather from its cytotoxicity.

*Acmella oleracea* L. (or *Spilanthes acmella*) is a medicinal plant renowned for its traditional use as a remedy with known anti-inflammatory, antioxidant, analgesic, and hepatoprotective properties [51]. The primary compound responsible for these effects is the alkylamide spilanthol, but the EO is known to contain *β*-pinene, myrcene, (E)-caryophyllene, and *α*-humulene [25]. By utilising an EO extracted from *A. oleracea* rich in spilanthol, researchers have investigated its cytotoxic effects and potential to regulate the responses of BV-2 microglial cells to LPS [25]. Both the EO and spilanthol deceased the cell viability in a concentration-dependent manner. Only the EO of *A. oleracea* was found to decrease the generation of reactive oxygen species in LPS-stimulated BV-2 cells (Table 1), indicating that the activity is not related to spilanthol. The inflammatory response was also attenuated, as evidenced by a reduction in the mRNA expression of iNOS, COX-2, IL-1*β*, and TNF-*α* [25].

### 4.3. Fabaceae Family Associated EOs

The oleoresin derived from Copaifera trees is used medicinally among indigenous communities in the Neotropics to address a range of medical conditions including microbial infections, inflammation, and the treatment of open wounds [52]. In recent years, the volatile fraction from oleoresin has been used in the cosmetic, food, and wellness industries [53]. The main constituent of the EO isolated from oleoresin (copaiba EO) is *β*-caryophyllene, and this compound is believed to be the activity carrier (i.e., the one responsible for the entire EO activity). In HMC3 microglial cells, a cell line derived from human microglia expressing AKT serine/threonine kinase 3 (Akt3), treatment with copaiba EO led to a transient increase in the phosphorylation of signalling proteins in the pI3K/Akt/mTOR cascade, which is associated with cell proliferation (Table 1) [54]. Thus, if the oil specifically impacts Akt3 activation [54], it may directly regulate immune system cell proliferation and signalling. In the same experimental setting, copaiba EO upregulated MAPK and JAK/STAT in HMC3 microglial cells [54], which directly links to the regulation of inflammatory response pathways in microglial cells [55].

### 4.4. Ericaceae Family Associated EOs

The genus Rhododendron belongs to the Ericaceae family and includes plants that are widely recognised for their anti-inflammatory properties in traditional medicine [27]. Essential oils of the flowers and leaves of *Rhododendron albiflorum* (cascade azalea) were found to differ in their content significantly. Namely, the EO obtained from the flowers was comprised mainly of monoterpenes (92%), while the EO obtained from leaves contained mostly sesquiterpenes (90.9%), with a small number of monoterpenes [27]. A recent study showed that this difference in EO composition influences the EO’s ability to inhibit Ca^2+^ signalling in C20 primary human microglial cells transformed into a clonal cell population (Table 1). Leaf EO rich in viridiflorol, spathulenol, curzerene, and germacrone inhibits intracellular Ca^2+^ mobilisation, and at the same time, inhibits Ca^2+^ influx from the extracellular space [27]. This observed activity may account for the ability of leaf EO to influence immune cell function, underlying the traditional utilisation of this plant as an anti-inflammatory medicinal source [27].

### 4.5. Cannabaceae Family Associated EOs

Non-psychotropic *Cannabis sativa* L. (i.e., industrial hemp) has several pharmacological properties due to a variety of active constituents, and the majority of them can be found in the EO. These include monoterpenes and sesquiterpenes, and sometimes, cannabinoids are also known to possess insecticidal, antimicrobial, fungicidal, antioxidant, the inhibition of acetyl-cholinesterase, and neuroprotective activities [56]. Three examined samples of different EOs obtained from varieties of *C. sativa* contained myrcene (20.3%) and *α*-pinene (8.5%)—oil 1; terpinolene (30.5%) and myrcene (11.1%)—oil 2; rich sesquiterpenes such as (E)-caryophyllene (18.9%), caryophyllene oxide (6.6%), and *α*-humulene (4.9%)—oil 3 [56]. In a series of in vitro tests, all three EOs failed to produce noticeable toxicity towards BV-2 cells, and only oils 2 and 3 in the concentration of 5 × 10−3 µL/mL inhibited the production of NO in the culture of BV-2 cells exposed to LPS [56]. In a subsequent experiment, the evaluated oils 2 and 3 were evaluated for their ability to alter the upregulation of pro-inflammatory genes (IL-1*β*, TNF-*α*, IL-6, COX-2, iNOS, and NLRP3) and downregulation of anti-inflammatory genes (IL-4 and MRC1) induced by LPS in BV-2 culture. Interestingly, only oil 3 significantly reduced the expression of all pro-inflammatory genes and significantly increased the expression of IL-4 [Barbalace]. When the pathways associated with the inflammatory cascade were investigated, it was shown that oil 3 prevented the translocation of NF-kB to the nucleus and strongly reduced p38 phosphorylation triggered by LPS, having at the same time no impact on the phosphorylation of Akt [56]. The examined EO 3 also prevented ROS generation and maintained reduced glutathione content in the LPS-exposed cells [56]. These results indicate that (E)-caryophyllene and its oxide and *α*-humulene might be the activity carriers, since the oil rich in these compounds and examined on their own produced strong anti-inflammatory activity in neuroglia cells [56].

### 4.6. Oleaceae Family Associated EOs

The flowers and flower buds of *Jasminum grandiflorum* L. have been used to prepare extracts to treat disorders such as hepatitis, stomatitis, and some psychiatric disorders throughout South Asia [57]. The flower EO of *J. grandifolium* containing ten different classes (monoterpenes, sesquiterpenes, diterpenes, aromatic alcohols, phenols, aromatic aldehydes, aromatic esters, aliphatic aldehydes, fatty alcohol, and alkanes) was found not to affect the cell viability and shape (grossly estimated) in a concentration span from 7.5 to 30 μg/mL [57]. At the same time, *J. grandifolium* EO decreased NO production and IL-1*β* and TNF-*α* (using Western blot) and prevented ROS generation within the cell, estimated as a decrease in cell culture fluorescence [57].

### 4.7. EOs Associated with Plants Used in Chinese Traditional Medicine

Chinese knowledge of botanic medicine dates back to around 3000 BC and is collected in Chinese materia medica. Selected plants are commonly utilised to create infusions, often combined with other plants in predetermined ratios. This practice results in specific formulations or decoctions tailored for treating particular illnesses and conditions [58].

*Cinnamomum camphora* L. Presl has been widely used in traditional Chinese medicine to treat inflammation and some muscle and joint degenerative states, and studies have shown their anti-inflammatory, antibacterial, antioxidant, antifungal, and repellent properties [59]. The investigated *C. camphora* EO, rich in linalool and with less abundance of eucalyptol, isoborneol, *α*-terpineol, and camphor, did not cause any toxic effect when present in the culture medium of BV-2 cells up to 250 µg/mL [59]. Exposure of BV-2 cells to LPS led to an upregulation in IL-6, IL-18, IL-1*β*, and TNF-*α* transcription and secretion, which was decreased in a concentration-dependent manner by the *C. camphora* EO [59]. Furthermore, metabolomics analysis revealed that *C. camphora* EO affects alanine, aspartate and glutamate metabolism, the tricarboxylic acid cycle, galactose metabolism, fatty acid biosynthesis, and pantothenate and coenzyme A (CoA) biosynthesis, among which the metabolism of amino acids and glucose metabolism (tricarboxylic acid cycle) were the most affected [59]. More profound molecular studies and associations between the alteration in metabolic pathways and inflammation signalling are needed to understand the exact connections and the role of the examined EO.

*Patrinia scabiosaefolia* is a perennial plant belonging to the Valerianaceae family, native to Eastern Asia [31]. The plant is traditionally used to treat different inflammatory-associated conditions. Their anti-inflammatory properties are partially attributed to triterpenoids, iridoids, flavonoids, and sterols [60]. Tested in concentrations up to 250 μg/mL EO of *P. scabiosaefolia* did not produce any notable cytotoxicity in several cell lines including BV-2 microglia [31]. Furthermore, the same EO tested at 100, 150, and 200 μg/mL prevented LPS-induced IL-1*β* and IL-6 production in the BV-2 cells (Table 1) [31].

*Schisandra chinensis* (Turcz.) Baill. is a woody plant native to Asia. Fruits from this plant are used as food (eaten fresh or in wine and tea) and in traditional Chinese medicine for menopause and pneumonia [33]. The active principles of *S. chinensis* mainly include lignans, essential oil constituents, polysaccharides, and terpenoids. S. chinensis EO at a concentration rising to 25 μg/mL decreased LPS-induced NO, IL-1*β*, IL-6, and TNF-*α*, but did not exhibit cytotoxicity in the BV-2 cell cultures (Table 1). Further molecular studies revealed that the EO prevented the nuclear translocation of NF-κB and the phosphorylation of MAPK in activated BV-2 cells [33]. Additionally, the p38 signalling pathway found to inhibit was found to be affected under similar experimental conditions. In a subsequent in vivo study, the S. chinensis EO was found to improve the cognitive performance of mice with A*β*1–42 or LPS-induced neurodegeneration and suppressed the expression of pro-inflammatory cytokines in hippocampal tissue [33]. Interestingly, *S. chinensis* EO inhibits p38 activation but has little effect on ERK1/2 and JNK signalling, which is not in accordance with the in vitro findings [33]. Such discordance between the in vitro and in vivo outcomes may arise from the differential ability of EO components to penetrate the BBB or due to metabolism following intragastric application.

*Thuja orientalis* L. is used to treat gout, rheumatism, diarrhoea, chronic tracheitis, hypertension, hematemesis, epistaxis, and haemorrhoids in Asia (Japan, Korea) [26]. This tree is rich in EO, containing majorly mono- and sesquiterpenoids such as *α*-pinene, Δ-3-carene, and *α*-cedrol [61]. Eleven sesquiterpenoids isolated from the EO of *T. orientalis* are found to inhibit NO production in LPS-stimulated BV2 microglia cells with IC50 values of 3.93–17.85 μM in the absence of cytotoxicity (Table 1). Some newly detected sesquiterpenoids, 3*α*-methoxy-4α-epoxythujopsane and Δ3,4-thujopsen-2,15-diol, are even more potent compared to a standard NO synthesis inhibitor, NG-monomethyl-L-arginine [26].

Ginger (*Zingiber officinale*, Zingiberaceae) has been used for over 2000 years as a food, spice, and medicinal source. The medicinal effects of ginger primarily arise from its active constituents, which include paradols, shogaols, and gingerols [62]. Ginger has been extensively studied as a potential neuroprotective and anti-ageing agent, protecting against inflammation and oxidative stress in neurodegenerative and age-related diseases [63]. The ability of ginger EO to prevent microglial activation in the presence of LPS was found to be mediated by TLR4 and downstream pathways such as NF-kB and MAPK [36]. TLR4 expression is, in part, known to be regulated by hypoxia-inducible factor-1 alpha (HIF-1*α*), which is, in turn, decreased by the EO of ginger [64]. One of ginger’s main EO constituents, ar-turmerone, was found to suppress the amyloid *β* (*Aβ*)-induced expression and activation of matrixmetaloproteinase 9 (MMP-9), iNOS, and COX-2 in microglia cells as well as to inhibit the phosphorylation and degradation of IκB-*α* and the phosphorylation of JNK and p38 MAPK [65]. A specific compound found in ginger, 6-gingerol (6-G), a compound contributing to the unique flavour and pharmaceutical value of ginger, has neuroprotective properties as reported in both in vitro and in vivo tests including an LPS-induced microglial inflammation model that involves the activation of BV-2 cells and the activation of a cascade of intracellular events (Table 1) [66]. To gain more insight into the neuroprotective effects of ginger and its mechanisms of action, Liu and co-workers [67] investigated the neuroprotective effects of gingerol in a model of induced hypoxia-reoxygenation in mouse neuroblast Neuro-2a cells. Gingerol ameliorated neuronal damage by regulating the miR-210/brain-derived neurotrophic factor (BDNF) axis [67], suggesting that gingerol may exert neuroprotective effects by modulating specific molecular pathways.

EO from a plant belonging to the Zingiberaceae family, *Alpinia officinarum* Hance, has also been investigated for its potential to influence microglial cells [32]. In traditional Chinese medicine, perennial rhizomes of this plant are used to cure stomach ache, relieve colds, invigorate the circulatory system, treat vomiting, and alleviate swelling. In the pharmacopoeia, it is suggested that for a plant to exert its beneficial properties, dried rhizomes should contain 1,8-cineole in more than 0.15% wt% [68]. EOs from different regions of China with varying contents of 1,8-cineole inhibit microglial BV-2 cell proliferation with IC_50_ values exceeding 200 μg/mL. The authors conclude that the main principle of the EO producing this effect is 1,8-cineole. They did not, however, exclude the possible impact of other EO constituents [32].

*Acorus tatarinowii* Schott is used to treat AD and its EO is considered the main activity carrier. Investigations on the effects of *A. tatarinowii* EO on neuroinflammation are relatively scarce. Traditional usage of this plant laid the foundation for an in vivo study of the EO on cognitive impairment in the AppSwe/PSEN1M146V/MAPTP301L triple transgenic (3 × Tg-AD) mouse model of AD [69]. The EO applied over eight weeks in doses of 50 and 100 mg/kg improved the animals’ cognitive performance in the Morris water maze and step-down avoidance tests. EO prevented neuronal damage by reducing A*β* and Tau phosphorylation and associated apoptosis driven by elevated Bax and reduced Bcl-2. Furthermore, the EO decreased both the content and expression of inflammatory mediators including IL-1*β*, TNF-*α*, IL-6, and IL-18 [69]. Finally, the authors concluded that the neuroprotective action of the EO arose from the inhibition of the nucleotide-binding domain and leucine-rich repeat protein 3 (NLRP3) and caspase-1 through inhibition of the NF-κB signalling pathway [69]. The major components of *A. tatarinowii* oil included *β*-asarone, *α*-asarone, methyl-eugenol, and caryophyllene, which are known to cross the BBB and exert a neuroprotective effect, as suggested elsewhere [70]. It has also been reported that stimulating the sense of smell with the volatile oil of *A. tatarinowii* can enhance the learning-memory performance of rats in a D-galactose and aluminium chloride-associated AD model [71].

*Valeriana jatamansi* Jones rhizome and radix, belonging to the Valerianaceae family, is indexed in the Chinese Pharmacopoeia (Part 1) for the treatment of mental disorders such as depression, anxiety, epilepsy, and insomnia [72]. The EO and compound-rich fractions isolated from *V. jatamansi* rhizoma et radix were evaluated in vitro for their ability to inhibit primary mouse microglial cell activation by LPS (Table 1). It was determined that different fractions exert a more or less pronounced effect on supressing IL-1*β* and IL-6 production [29]. There has been an established relationship between the chemical composition of the EO from *V. jatamansi* rhizoma et radix in inhibiting microglial activation. The major components primarily responsible for inhibiting microglial activation were found to be vetivenol, bornyl acetate, seychellene, and *β*-elemene [29].

The EO derived from *Pterodon emarginatus* Vogel seeds, which includes *β*-elemene and *β*-caryophyllene sesquiterpenes, has shown promise in mitigating the neurological symptoms and progression of autoimmune encephalomyelitis in C57BL/6 mice [30]. The observed effects can be attributed to the modulation of the Th1/Treg immune balance during the 25 d experiment. Specifically, there is a reduction in the Th1 cell-mediated immune response (CD4+ T cells associated with inflammation) and an enhancement in the Treg response through an increase in IL-10 production. Moreover, the EO inhibited both microglial activation and the expression of iNOS associated with the inhibition of axonal demyelination and neuronal death [30]. The results of this study provide a rationale for using *P. emarginatus* in treating peripheral and central autoimmune disorders in keeping with its medicinal properties and use in folk medicine.

### 4.8. EO Associated with Plants Used as Foodstuff

Carrot (*Daucus carota*) EO has diuretic, antimicrobial, and anti-inflammatory effects [73]. The potential of *Daucus carota* subsp. *maximus* EO in preventing N9 mouse microglial cell inflammatory response by determining NO content after exposure to LPS was recently evaluated. This EO in a concentration of 0.32 μL/mL reduced NO production (Table 1) in the cells by 35.8% [73]. There was no notable cytotoxicity at the concentrations required to produce these effects. At a twofold higher concentration, the viability of N9 cells was reduced by 30% [73]. Analysis of the EO found that the most dominant constituents were *α*-pinene, geranyl acetate, β-bisabolene, α-asarone, and E-methylisoeugenol, among which *α*-pinene and geranyl acetate were deemed to be the constituents responsible for activity [73].

*Angelica dahurica* is a plant species widely cultivated and native to various regions including Siberia, the Russian Far East, Mongolia, North-Eastern China, Japan, Korea, and Taiwan. This plant is commonly found in diverse environments, typically near riverbanks, along streams, and among rocky shrubs [74]. In traditional medicine, it is used for managing headaches, relieving nasal obstruction, as a pain reliever, anti-inflammatory drug, laxative, sedative, and to “purge the body” overall [74]. The stalks of this plant are commonly utilised as a food ingredient, while the seeds serve as seasoning condiments in food, imparting flavour, etc. [74]. When the EO of *A. dahurica* was tested in BV-2 cell culture, it exerted moderate cytotoxicity, with IC_50_ values at concentrations higher than 193 μg/mL [34].

*Myristica fragrans* Houtt. is a well-renowned evergreen tree with fruit seed that serves as a source of nutmeg [75]. It is native to Indonesia, however, it can be found in Grenada, India, Sri Lanka, Mauritius, South Africa, and some parts of the USA. In traditional Ayurvedic medicine, *M. fragrans* has been used to treat anxiety, nausea, diarrhoea, cholera, stomach cramps, paralysis, and rheumatism [75]. The EO of *M. fragrans* mainly contains monoterpenes (sabinene, *β*-pinene and *β*-terpineol), phenylpropene (eugenol and myristicin), and sesquiterpenes (germacrene D and *β*-bergamotene) [75]. The EO of *M. fragrans* tested in parallel with other EOs from different medicinal plants exerted the lowest toxicity towards BV-2 microglial cells [34].

*Pelargonium graveolens* or rose geranium belongs to the Geraniaceae family, which is extensively used as foodstuff (cakes, jams, jellies, ice creams, sorbets, salads, etc.) in the perfume industry and for aromatherapy [76]. *Pelargonium graveolens* (geranium oil) was evaluated for its impact on the inflammatory cytokine production of BV-2 microglial cells after exposure to LPS, where the EO inhibited the production of NO and the expression of COX-2 and iNOS (Table 1) [77]. Activity of the EO was attributed to citronellol, citronellyl formate, linalool, geraniol, isomenthone, and menthone, which made up to 60% of the total oil constituents [77]. The mentioned compounds, on their own, poorly affected NO production, except citronellol. Thus, the net EO action was due to their synergistic action [77].

In some instances, foodstuffs could be a valuable source of EO valued for medicinal properties, with the name given to this concept as “medicine food homology” [34]. In this context, herbs are valued for their potential therapeutic properties in traditional medicine and are considered beneficial for maintaining health when incorporated into the diet. This duality highlights the belief that certain herbs, beyond their medicinal applications, can contribute to health preservation when consumed as part of a regular diet or when added as functional food ingredients.

**Table 1 ijms-25-05168-t001:** A summary of the anti-inflammatory like properties of essential oils in the stimulated microglial cells.

Essential Oil/Plant Part	Experimental Setting	Observed Effect	Mechanism of Action	References
*Acmella oleracea*/aerial parts/inflorescence	LPS-activated BV-2 microglia	Decreased iNOS, COX-2, IL-1*β*, and TNF-*α* mRNA expression.	NA	[25]
*Artemsia dracunculus*/aerial parts	LPS-activated BV-2 microglia	Increased catalase and superoxide dismutase.	NA	[24]
*Artemisia herba-alba*/aerial parts	LPS-activated BV-2 microglia	Decreased NO production.	NA	[23]
*Cannabis sativa* L./aerial parts/inflorescence	LPS-activated BV-2 microglia	Decreased NO, iNOS, COX-2, NLRP3, IL-6, IL-1*β* and TNF-*α*, increases IL-4 Reduced NF-kB translocation and phosphorylation of p38, did not affect Akt phosphorylation.	NF-κB p38	[56]
*Cinnamomum camphora* L./leaves	LPS-activated BV-2 microglia	Decreased IL-6, IL-18, IL-1*β*, and TNF-*α* secretion and mRNA expression.	Affects amino acid metabolism (alanine and aspartate) and tricarboxylic acid cycle	[59]
*Copaifera* sp./oil resin	HMC3 microglial cells	Phosphorylation of pI3K/Akt/mTOR positive regulation of MAPK and JAK/STAT signalling.	pI3K/Akt/mTOR MPAK JAK/STAT	[54]
*Daucus carota* subsp. *Maximus*/aerial parts	LPS-activated N9 microglia	Decreased NO production.	NA	[73]
*Jasminum grandiflorum* L./flowers	LPS-activated BV-2 microglia	Decreased in NO, IL-1*β*, and TNF-*α*, decrease in ROS production.	NA	[57]
*Pelargonium graveolens*/whole plant	LPS-activated BV-2 microglia	Decreased NO production.	Decreases iNOS and COX-2	[77]
*Patrinia scabiosaefolia*/whole plant	LPS-activated BV-2 microglia	Decreased in IL-1*β* and IL-6.	NA	[31]
*Rhododendron albiflorum*/flowers and leaves	LPS-activated C20 microglia	Inhibited Ca^2+^ signalling.	Inhibits Ca^2+^ mobilisation and influx	[27]
*Schisandra chinensis*/NA	LPS-activated BV-2 microglia	Decreased in NO, IL-1*β*, IL-6 and TNF-*α* concentration.	NF-κB MPAK p38/JNK/ERK	[33]
*Thymus vulgaris*/NA	LPS-activated BV-2 microglia	Decreased TNF-*α* and IL-6 mRNA expression.	NF-κB C/EBP*β*	[37]
*Thymus zygis* L. subsp. *sylvestris*/aerial parts /inflorescence	LPS-activated BV-2 microglia	Decreased NO production.	NA	[28]
*Thuja orientalis*/aerial parts	LPS-activated BV-2 microglia	Decreased NO production.	NA	[26]
*Valeriana jatamansi* Jones Rhizoma et Radix/rhizome and root	LPS-activated primary mouse microglia	Decreased in IL-1*β* and IL-6 concentration and mRNA.	NA	[29]
*Zingiber officinale*/NA	LPS-activated BV-2 microglia	Decreased TNF-*α* and IL-1*β*, NO, MMP-9, PGE2.	TLR4, NF-κB, MAPK, HIF-1*α*	[36,65,67]
*Ziziphora clinopodioides*/aerial parts/inflorescence	Wild type BV-2 and BV-2 (Acox1−/−)	Decreased in catalase.	NA	[46]

Acox1−/−—Acyl-CoA oxidase 1-deficient BV-2 cells; COX-2—cyclooxygenase 2; HIF-1α—hypoxia-inducible factor-1 alpha; IL-6—interleukin 6; MAPK—mitogen-activated protein kinase; MMP-9—matrixmetaloprotease-9; NO—nitric oxide; PGE2—prostaglandin E2; TLR—Toll-like receptor; TNF-*α*—tumour necrosis factor α; mRNA—micro ribonucleic acid; NA—not applicable.

## 5. Proposed Anti-Inflammatory Mechanisms of Selected Essential Oil Constituents

The specific pathways through which EOs modulate microglia and associated inflammation are currently under continuous investigation. Ongoing research suggests that various EOs and their components may target distinct pathways, leading to diverse effects [37]. Some studies have reported anti-inflammatory and neuroprotective actions associated with these mechanisms. In the following section, the main constituents present in EOs responsible for these EO actions are further considered.

1,8-Cineole (Figure 3), the primary active compound in *R. officinalis*, *E. globulus,* and *A. officinarum* EOs, engages several pathways affecting microglial cell growth, proliferation, and inflammatory response. 1,8-Cineole reduced the proinflammatory cytokine levels including TNF-*α*, IL-1*β*, and IL-6 in the A*β*-treated BV-2 cells and also downregulated the expression of NOS-2, COX-2, and NF-κB, making it a potential candidate treatment for inflammation associated with A*β* 28a. In LPS-exposed human umbilical vein endothelial cells (HUVECs), the expression of PPAR-γ was also increased by 1,8-cineole. This compound also decreased vascular cell adhesion molecule-1 (VCAM-1), IL-6, and IL-8 protein, and mRNA levels in LPS-exposed HUVECs. GW9662, an inhibitor of PPAR-γ, could counteract these effects, indicating that the CPPAR-γ signalling pathway is involved in its action [78].

*β*-Caryophyllene (Figure 3) is a bicyclic sesquiterpene, which mainly occurs in the form of trans-caryophyllene and in combination with small amounts of its isomers (iso-caryophyllene and *α*-caryophyllene or *α*-humulene), but also as its oxidative derivative *β*-caryophyllene oxide [79]. This sesquiterpene is present in numerous EOs such as *Ocimum basilicum* L., *Cinnamomum* sp., *Lavandula angustifolia* Mill., *Origanum vulgare* L., and *Rosmarinus officinalis* L. [79]. The effects of *β*-caryophyllene are mediated by activating the cannabinoid pathway via cannabinoid receptor type 2 (CB2), preventing dopaminergic neuron damage in an in vivo model by lessening the activation of microglial cells [80]. Direct modulation of the neuroinflammatory response of *β*-caryophyllene modulates inflammatory responses via TLR, and associated MAPK and ERK signalling [69,81]. Additionally, in both rat and mice models of CNS degeneration, *β*-caryophyllene modulated the inflammatory response via CB2 receptor activation [82] and the PPAR-γ signalling pathway and associated MAPK/ ERK signalling [69].

Geraniol (Figure 3), an acyclic monoterpene alcohol found in the EO of plants such as geranium, lemongrass, and rose, targets several signalling cascades. In BV-2 microglia stimulated with LPS, geraniol prevented TNF-*α* mRNA expression and to a lesser extent, IL-6 mRNA expression. This was achieved through inhibition of the NF-kB pathway and decreased chromatin-bound phosphorylated C/EBP*β* [37]. In an in vivo experiment with high-fat diet-induced brain damage and ageing in rats, geraniol was found to produce significant anti-inflammatory (decrease in IL-1*β*, iNOS, NF-κBp65, and COX-2 expression) and antioxidant (boosting neuronal reduced glutathione content, CAT, SOD, and decreasing thiobarbituric acid reactive substances, NO, and xanthine oxidase) activity, which in turn was associated with an increased capability for learning and memory [82]. Additionally, geraniol prevented spinal cord injury in an animal model by alleviating the inflammatory response through the NF-κB and p38MAPK pathways, decreasing apoptosis and oxidative damage [83]. In the Ox-LDL-stimulated HUVECs cells, mimicking vascular cell injury associated with numerous CNS disorders geraniol inhibited the nuclear translocation and activity of NF-κB and the phosphorylation of IkB*α* [84]. Furthermore, geraniol activates the PI3K/AKT/NRF2 pathway in HUVECs, increasing HO-1 expression. Its inhibitory effects on Ox-LDL-induced inflammation and oxidative stress involve targeting the PI3/AKT/NRF2 pathway in HUVECs [84].

Menthol (Figure 3), a primary EO component derived from plants belonging to the *Mentha* species, has been suggested to protect against neuroinflammation both in vitro and in vivo [85]. Tested in a concentration-dependent range from 2.5 to 40 μmol/L, menthol did not exert any toxicity towards BV-2 cells. In subsequent experiments, menthol inhibited the upregulation of iNOS and COX-2 mRNA, and IL-1*β*, IL-6, and TNF-*α* mRNA [85]. In the same experimental setting, menthol prevented the LPS-induced phosphorylation of AKT, p65, JNK1/2, and ERK1/2, but not p38, suggesting that it prevents the upregulation of the aforementioned cytokines by affecting the NF-κB and MAPK signalling pathways [85]. Furthermore, in an in vivo study, menthol’s effects contributed to the recovery of damaged dopaminergic neurons and motor function in a rat model of LPS-induced Parkinson’s disease [84]. The action of menthol in this study and in a study performed with the *Menthae* sp. extract [86] showed neuroprotective effects, pointing to Mentha species and menthol possessing anti-inflammatory properties.

Linalool (Figure 3), a well-known compound present in the EOs of plants such as *T. vulgaris*, *Ocimum* spp., *Ziziphora* spp., *A. dracunculus* L., *P. cablin*, *A. oleracea*, and *Lavandula angustifolia*, etc. has been relatively well-studied, and its activity can be observed through three target mechanisms within microglia cells. First, the anti-inflammatory action of linalool was observed in microglia cells exposed to LPS. The Nrf2 pathway mediated the effect of linalool in cultured BV2 microglia cells exposed to LPS, thus producing noticeable concentration-dependent anti-inflammatory action [87]. More precisely, linalool causes the translocation of Nrf2 to the nucleus, leading to the increased production of HO-1 and, consequently, to a decrease in inflammatory mediator (TNF-*α*, IL-1*β*, NO, and PGE2) production through the inhibition of gene expression [35]. Additionally, linalool decreases NF-kB signalling pathway activation by reducing chromatin-bound active p50 and p65 [37]. Second, linalool decreases ROS production by directly influencing free radical generation, increasing enzymatic antioxidant capacities (e.g., glutathione peroxidase) or through its scavenging properties [35,88]. Finally, linalool, as the main constituent of the *Coriandrum sativum* (70%), was found to be potentially associated with delayed cerebral amyloidosis including the accumulation of amyloid deposits and *β*-amyloid peptides in the hippocampal tissue of ageing triple transgenic state mutation mice through the reduction in ROS and a decrease in IL-1*β*, iNOS, and COX-2 [35].

*β*-Ionone (Figure 3), an analogue of *β*-carotenoid and with vitamin A-like activity present in many EOs, is known for its polypharmacological activity including antiapoptotic, tumour suppressing, chemopreventive (the ability of a substance to prevent cancer (re)occurrence), and antioxidant [89] actions. Therefore, there is a potential of *β*-ionone in preventing the activation of BV-2 microglia cells treated with LPS. Mechanistic studies showed that *β*-ionone decreases mRNA for iNOS and COX2, a consequence of reduction in the DNA-binding activity of NF-kB through the suppression of the nuclear translocation of p50 and p65 [89]. Additionally, the authors suggest that *β*-ionone might exert anti-inflammatory activity in this model by affecting MAPK, but further studies are needed [90].

*β*-Patchoulene (*β*-PAE) (Figure 3), a major active compound found in *Pogostemon cablin* (patchouli oil), displays anti-inflammatory-like properties on RAW 264.7 macrophages following stimulation with LPS. This effect is attributed to *β*-PAE’s ability to balance the production of pro-inflammatory and anti-inflammatory cytokines. Prior treatment with *β*-PAE led to a reduction in the synthesis of TNF-*α*, IL-6, and IL-1*β*, along with an increase in IL-10 expression. Moreover, *β*-PAE effectively inhibited signalling via iNOS and COX-2, resulting in decreased levels of NO and PGE2, respectively. The activation and translocation of NF-κB play a central role in the onset and progression of acute inflammation, as it is necessary for the transcription of numerous pro-inflammatory mediators. *β*-PAE hampers the translocation of NF-κB from the cytoplasm to the nucleus and stabilises the cytoplasmic nuclear factor-IκB*α* complex [90].

Thymol, a colourless crystalline monoterpene phenol, holds significant dietary importance as a key constituent in various thyme species [91]. The neuroprotective activity of thymol was not examined in in vitro models; however, it was investigated in several in vivo models, proving its efficacy. Thymol prevented neurotoxicity and neurodegeneration in a rat model of rotenone-induced neurodegeneration. These effects were attributed to preserving endogenous antioxidant defence networks and reducing inflammatory mediators including cytokines and enzymes (CAT and SOD) [92]. Additionally, thymol reduced neuronal damage and memory impairment following intrahippocampal A*β* peptide injection in rats fed with a high-fat diet [93]. Application of thymol is known to decrease hippocampal pro-inflammatory cytokines (TNF-*α* and IL-1*β*) released from astrocytes and microglia during and after seizure induction in rats [91].

## 6. Conclusions

Previously, anti-inflammatory and neuroprotective effects of different EOs have been shown, however, detailed mechanistic studies on their activity are insufficient. Depending on the EO composition, the observed activity varies, and may involve multiple molecular mechanisms in suppressing microglial function, inflammation, and neurodegeneration. Some of the most potent and promising EO constituents include 1,8-cineole, *β*-caryophyllene, geraniol, menthol, linalool, *β*-ionone, *β*-patchoulene, and thymol. These have been studied both in vitro and to a more limited extent in vivo, proving their effectiveness. However, further studies to confirm these findings are warranted.

The underlying mechanisms of EOs in modulating microglial and microglial-associated inflammation involve the modulation of signalling pathways, the inhibition of microglial cell activation, regulation of the expression and production of inflammatory mediators, antioxidant effects, and potential modulation of the gut–brain connection. These mechanisms collectively contribute to the anti-inflammatory and neuroprotective properties of EOs. However, more research is needed to verify the specific molecular pathways involved. This review points to the most investigated EO constituents, which include 1,8-cineole, geraniol, and linalool, providing information that aids efforts in the development of EO components as treatments for neuroinflammatory and degenerative diseases where microglial-associated inflammation is a contributing mechanism.

Finally, more in-depth studies are needed to better understand the actions of EOs and their constituents. These studies should investigate other signalling pathways and the interaction of different cells in the inflammation process. Finally, in vivo studies providing a conformation of the activity observed under in vitro conditions should be conducted.

## Figures and Tables

**Figure 1 ijms-25-05168-f001:**
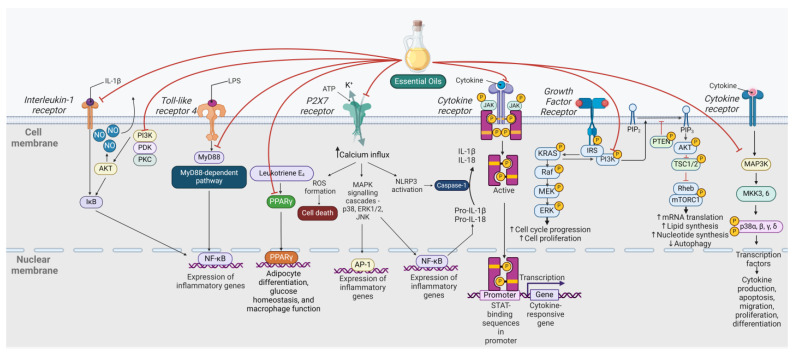
Cell signalling pathways modulated by essential oils. Abbreviations: AKT, protein kinase B; ATP, adenosine triphosphate; ERK, extracellular signal-regulated kinase; IκB, inhibitor of kappa B; IRS, insulin receptor substrate; JAK, Janus kinase; JNK, c-Jun N-terminal kinase; KRAS, Kirsten rat sarcoma viral oncogene homolog; MAPK, mitogen-activated protein kinase; MEK, mitogen-activated protein kinase kinase; mTORC, mammalian target of rapamycin complex; MyD88, myeloid differentiation primary response 88; NO, nitric oxide; NF-κB, nuclear factor-kappa B; PI3K, phosphoinositide 3-kinase; PKC, protein kinase C; PPAR, peroxisome proliferator-activated receptor; PTEN, phosphatase and tensin homolog; Raf, rapidly accelerated fibrosarcoma; Rheb, Ras homolog enriched in brain; ROS, reactive oxygen species; STAT, signal transducer and activator of transcription; TSC, tuberous sclerosis complex. Up arrows indicate increased activity and down arrows indicate decreased activity of the process. Created with Biorender.com, accessed on 23 April 2024.

**Figure 2 ijms-25-05168-f002:**
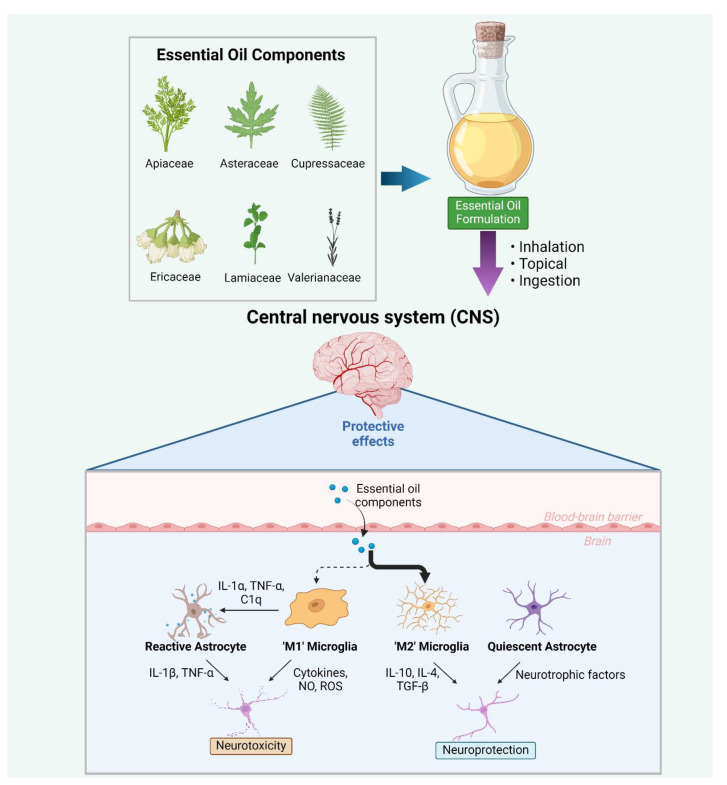
Anti-inflammatory and neuroprotective mechanisms of essential oils in the CNS. Essential oils of plants from the families of Apiaceae, Asteraceae, Cupressaceae, Ericaceae, Lamiaceae, Valerianaceae, and others have been shown to regulate microglia and offer therapeutic potential for CNS disorders where inflammation features. Components of essential oils have anti-inflammatory properties that shift microglial activation from the activated ‘M1’ phenotype towards the protective ‘M2’ microglial phenotype, which secretes neuroprotective factors such as IL-10, IL-4, and TGF-*β*. ‘M1’ type microglia, on the other hand, secrete factors such as nitric oxide and reactive oxygen species, which are toxic to neuronal cells. Furthermore, activated microglia-derived factors such as IL-1*α*, TNF-*α*, and C1q can induce a reactive astrocyte phenotype that have reduced capacity to promote neuronal survival and regulate synapse formation. Abbreviations: NO, nitric oxide; ROS, reactive oxygen species; TNF, tumour necrosis factor; TGF, transforming growth factor. Created with Biorender.com, accessed on 15 April 2024.

**Figure 3 ijms-25-05168-f003:**
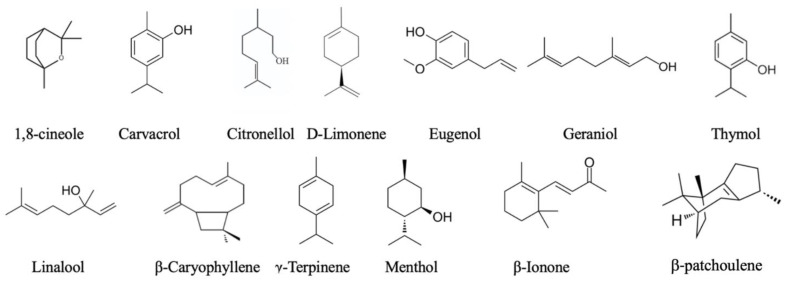
Chemical structure of EO constituents primarily responsible for anti-inflammatory actions.

## Data Availability

Data can be made available upon reasonable request from the corresponding author.

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
