# Peer review of "Essential Oil Constituents as Anti-Inflammatory and Neuroprotective Agents: An Insight through Microglia Modulation"

_ijms, 2024, doi:10.3390/ijms25105168_

Round 1
Reviewer 1 Report
Comments and Suggestions for Authors
This review manuscript encompasses some essential oils, as well as some of their components, and describes the evidence for their potential modulation of microglia and their potential pharmacological mechanisms.
In general, the authors should include their points of view including a comparison, critique and assessment of the studies reviewing.
What is EOs advance in comparison with some other natural product, food component, or phytochemical? The scientific reason for this review paper should be explained in more detail.
“Proposed anti-inflammatory mechanisms of selected essential oil constituents” should be presented in Table also.
Author Response
Please see the answer to your comments attached and below.
This review manuscript encompasses some essential oils, as well as some of their components, and describes the evidence for their potential modulation of microglia and their potential pharmacological mechanisms.
In general, the authors should include their points of view including a comparison, critique and assessment of the studies reviewing.
Answer: Thank you for your comment. The authors tried to add their comments, where appropriate, however, having in mind the diversity of the EO constituents, there is a narrow window for detailed meaningful comparison. We have added a general assessment of the reviewed studies in the conclusion section.
What is EOs advance in comparison with some other natural product, food component, or phytochemical? The scientific reason for this review paper should be explained in more detail.
Answer: Thank you for your suggestion. We have added in the introductory part, and throughout the manuscript the background of the EO and their better role in preventing CNS damage than other extracts.
“Proposed anti-inflammatory mechanisms of selected essential oil constituents” should be presented in Table also.
Answer: Similar results are given under the column entitled ‘Mechanism of action’. This is collected form the publications where the authors concluded that such activity was observed due to a specific and known reason. For every other case NA was added suggesting that the authors did not know what the underlining mechanism is, meaning that it could be due to numerous factors, which are sometimes hard to predict at all. Also, one should bear in mind that the definition of the mechanism of action is just based on a single pathway examination, and not on overlapping ones and non-specific ones. Thus, we withhold ourselves from suggesting/proposing mechanisms of action avoiding into making early and unsupported conclusions, which in this case might be misleading for future researchers having in mind that this is a review paper.

Reviewer 2 Report
Comments and Suggestions for Authors
I reviewed the manuscript entitled Essential Oil Constituents as Microglia Modulators: Explora tory Analysis on Mechanism of Action of Anti-inflammatory and Neuroprotective Effects.
I agree to accept this manuscript after major revision.
1) Abstract: β-caryophyllene, β-patchoulene carvacrol, β-ionon, all Greek letters require italics. Please make modifications to the entire text. β-patchoulene carvacrol, there should be a comma between two components.
2) Keywords: essential oils; Essential oils is correct.
3) reactive oxygen species [2]. This should be changed to reactive oxygen species (ROS), because this is where it first appeared. Afterwards, you can directly use ROS.
4) A literature survey using the Scopus search browser, including the keywords “essential oils’’ and “microglia’’ together (from 1990- May 2023), can the author obtain comprehensive and complete results using only Scopus database? I am worried. Can the author use databases such as Pubmed, Web of Science, Scifinder, etc. to query? See if there are any new results and gains. Also, it is already April 2024, can the search date be extended to this day? This way, it can ensure the timeliness of this article.
5) Detailed analysis has been conducted on several plant species belonging to genera such as Artemisia, Polygonum, Thymus, Zingiber and others in order to evaluate the anti-in-flammatory, genus names need to be italicized.
6) in vitro, in vivo, they all require italics. Check and modify the entire text.
7) During the first 24 hours of exposure cytotoxicity in both cell lines was observed when cells were exposed to 50 μl/ml of the EO. With prolonged cultivation of up to 72h in the presence of 0.5 μl/ml of EO, 24 hours should change to 24 h, do not use hours, instead use h because it is an international unit. Similarly, 50 μl/ml should be changed to 50μL/mL; 72h should be changed to 72 h. The above issues exist throughout the entire article and need to be carefully checked and corrected.
8) The EO of A. dracunculus consists of several main components such as estragole, also known as methyl chavicol or p-allyl anisole (concentrations of 40–85%), sabinene (35%), methyl eugenol (25%), and elemicin (57%). Is it the content of ingredients in different reports? Because the combined content of these components exceeds 100%, please verify.
9) Namely, the EO obtained from flowers was comprised mainly of monoterpenes (92%), while the EO obtained from flowers contained mostly sesquiterpenes (90.9%), with a small number of monoterpenes [27]. There should be an EO from leaf, please verify and modify it.
10) immune balance during the 25-day experiment. 25 d is correct.
11) Table 1. Copaiba , missing the name of the species.
12) decreasing thiobarbituric acid reactive substances (TBARS), it is not necessary to use abbreviations if they appear only once, and abbreviations are only necessary if they appear more than three times. Excessive abbreviations can confuse readers.
13) NF kB is incorrect, it should be NF-kB.
14) I have read all the references and found some issues. Ref 72, 2021;10(4):546. The year needs to be bolded, and a comma should be used after the issue. ref 78, 327–352 there should be a period after it. Ref 88 , 91 and 96, They all lack article numbers. The reference format needs to be unified, Please carefully check and make modifications.
15) The main question addressed by the research is using the Scopus search browser. Although this database is commonly used, it usually requires searching 3-5 databases to obtain comprehensive results. Therefore, I suggest the author to supplement relevant searches.
16) The discussion in this article is relatively systematic and complete, the conclusion is consistent with the evidence and arguments provided.
Author Response
Please see the answer to your comments attached and below.
I reviewed the manuscript entitled Essential Oil Constituents as Microglia Modulators: Explora tory Analysis on Mechanism of Action of Anti-inflammatory and Neuroprotective Effects.
I agree to accept this manuscript after major revision.
1) Abstract: β-caryophyllene, β-patchoulene carvacrol, β-ionon, all Greek letters require italics. Please make modifications to the entire text. β-patchoulene carvacrol, there should be a comma between two components.
Answer: Thank you for noticing the mistakes, corrections were made as suggested throughout the manuscript.
2) Keywords: essential oils; Essential oils is correct.
Answer: Thank you for noticing the mistake, corrections were made as suggested.
3) reactive oxygen species [2]. This should be changed to reactive oxygen species (ROS), because this is where it first appeared. Afterwards, you can directly use ROS.
Answer: Thank you for noticing the mistake, corrections were made as suggested.
4) A literature survey using the Scopus search browser, including the keywords “essential oils’’ and “microglia’’ together (from 1990- May 2023), can the author obtain comprehensive and complete results using only Scopus database? I am worried. Can the author use databases such as Pubmed, Web of Science, Scifinder, etc. to query? See if there are any new results and gains. Also, it is already April 2024, can the search date be extended to this day? This way, it can ensure the timeliness of this article.
Answer: Indeed, for our search only the Scopus database was used since it was proven in numerous occasions that it gives the best research output. Regarding the period of search, it was as stated 1990-May 2023, and after that an extensive work was performed in filtering the papers, detailed analysis of the selected ones, and writing of the paper. Thus, a final product of this work was around January 2024, and afterwards it was submitted. In order to encompass potential data from other databases we included Pubmed and SciFinder with the same filters, and we found only a few extra papers, all published after our initial search was done. All extra papers found were included in the manuscript.
5) Detailed analysis has been conducted on several plant species belonging to genera such as Artemisia, Polygonum, Thymus, Zingiber and others in order to evaluate the anti-in-flammatory, genus names need to be italicized.
Answer: Thank you for noticing the mistake, corrections were made as suggested.
6) in vitro, in vivo, they all require italics. Check and modify the entire text.
Answer: Thank you for noticing the mistakes, corrections were made as suggested.
7) During the first 24 hours of exposure cytotoxicity in both cell lines was observed when cells were exposed to 50 μl/ml of the EO. With prolonged cultivation of up to 72h in the presence of 0.5 μl/ml of EO, 24 hours should change to 24 h, do not use hours, instead use h because it is an international unit. Similarly, 50 μl/ml should be changed to 50μL/mL; 72h should be changed to 72 h. The above issues exist throughout the entire article and need to be carefully checked and corrected.
Answer: Thank you for noticing the mistakes, corrections were made as suggested throughout the manuscript.
8) The EO of A. dracunculus consists of several main components such as estragole, also known as methyl chavicol or p-allyl anisole (concentrations of 40–85%), sabinene (35%), methyl eugenol (25%), and elemicin (57%). Is it the content of ingredients in different reports? Because the combined content of these components exceeds 100%, please verify.
Answer: The authors understand why this was a bit confusing. We have rewritten this sentence and added a new one directly explaining the chemotype of the examined EO.
9) Namely, the EO obtained from flowers was comprised mainly of monoterpenes (92%), while the EO obtained from flowers contained mostly sesquiterpenes (90.9%), with a small number of monoterpenes [27]. There should be an EO from leaf, please verify and modify it.
Answer: Thank you for noticing the mistake, corrections were made as suggested.
10) immune balance during the 25-day experiment. 25 d is correct.
Answer: Corrections were made as suggested.
11) Table 1. Copaiba , missing the name of the species.
Answer: Since the Copaiba oil is obtained for oilresins from different trees of the Copaifera family, we added Copaifera sp.
12) decreasing thiobarbituric acid reactive substances (TBARS), it is not necessary to use abbreviations if they appear only once, and abbreviations are only necessary if they appear more than three times. Excessive abbreviations can confuse readers.
Answer: Corrections were made as suggested.
13) NF kB is incorrect, it should be NF-kB.
Answer: Corrections were made as suggested.
14) I have read all the references and found some issues. Ref 72, 2021;10(4):546. The year needs to be bolded, and a comma should be used after the issue. ref 78, 327–352 there should be a period after it. Ref 88 , 91 and 96, They all lack article numbers. The reference format needs to be unified, Please carefully check and make modifications.
Answer: Corrections were made as suggested.
15) The main question addressed by the research is using the Scopus search browser. Although this database is commonly used, it usually requires searching 3-5 databases to obtain comprehensive results. Therefore, I suggest the author to supplement relevant searches.
Answer: As suggested additional database search were performed and added.
16) The discussion in this article is relatively systematic and complete, the conclusion is consistent with the evidence and arguments provided.
Answer: Thank you for your comment.

Reviewer 3 Report
Comments and Suggestions for Authors
The review entitled “Essential Oil Constituents as Microglia Modulators: Exploratory Analysis on Mechanism of Action of Anti-inflammatory and Neuroprotective Effects” aims to clarify the molecular mechanisms underlying the effects of several essential oil components in the mechanism of microglial activation.
It is extremely interesting how essential oils derived from a wide variety of plants (e.g. Apiaceae, Asteraceae, Cupressaceae, Ericaceae, Lamiaceae, Valerianaceae and others) may modulate the neuroinflammatory response in the central nervous system, regulating the microglial activation ad the consequent production of anti-inflammatory cytokines, but also the release of neuroprotective molecules by glial cells (e.g. astrocyte).
The authors provide important insights into the possible study of these molecular mechanisms in various diseases and therapeutic strategies, from neurodegenerative diseases to cancer therapy.
The conclusions are well outlined and concise, clearly expressing the limitations of the proposed study.
The references reported are appropriate and include recent literature data, in line with the present study.
The following are just suggestions to better present the proposed work:
- The references section should be made uniform (see references 7, 41, etc.)
- Figure 1 is extremely complex and comprehensive. However, some molecular mechanisms depicted are too small and difficult to read. They should be slightly enlarged to facilitate the reader.
- The title seems to be too entangled, I would suggest making it more streamlined and impactful
Author Response
Please see the answers to yur comments attached and below.
The review entitled “Essential Oil Constituents as Microglia Modulators: Exploratory Analysis on Mechanism of Action of Anti-inflammatory and Neuroprotective Effects” aims to clarify the molecular mechanisms underlying the effects of several essential oil components in the mechanism of microglial activation.
It is extremely interesting how essential oils derived from a wide variety of plants (e.g. Apiaceae, Asteraceae, Cupressaceae, Ericaceae, Lamiaceae, Valerianaceae and others) may modulate the neuroinflammatory response in the central nervous system, regulating the microglial activation ad the consequent production of anti-inflammatory cytokines, but also the release of neuroprotective molecules by glial cells (e.g. astrocyte).
The authors provide important insights into the possible study of these molecular mechanisms in various diseases and therapeutic strategies, from neurodegenerative diseases to cancer therapy.
The conclusions are well outlined and concise, clearly expressing the limitations of the proposed study.
The references reported are appropriate and include recent literature data, in line with the present study.
The following are just suggestions to better present the proposed work:
- The references section should be made uniform (see references 7, 41, etc.)
Answer: Thank you for your comment, this has been corrected now.
- Figure 1 is extremely complex and comprehensive. However, some molecular mechanisms depicted are too small and difficult to read. They should be slightly enlarged to facilitate the reader.
Answer: Thank you for your comment, we have corrected the Figure.
- The title seems to be too entangled, I would suggest making it more streamlined and impactful
Answer: A new title has been suggested ‘Essential Oil Constituents as Anti-inflammatory and Neuroprotective agents: An insight through Microglia Modulation’.

Reviewer 4 Report
Comments and Suggestions for Authors
This is a quite well-written manuscript devoted to the neuroprotective activity of different essential oils. The topic is interesting.
However, the manuscript needs some improvements as listed below:
1) The aim of the review article should be clearly stated in the Introduction.
2) The methodology used for preparing the review (lines 119-123) should be placed in the Introduction. "Papers outside the scope of this review, found by this search because certain keywords were mentioned in the text, were excluded." - the exclusion criteria should be specified in detail.
3) Figures should be placed as near to their first mention in the text as possible to facilitate easier following
4) The quality of Figure 1 should be improved. Fonts should be larger because they are hardly visible.
5) Section 4 should be divided into subsections for clearer organization (e.g., taking into account families). Moreover, in my opinion, there is too much unnecessary information in this section (see e.g., lines 182-193), and sometimes information directly linked with the aim of the study is too sparse (see e.g., lines 199-202). Some parts are rich in details (e.g., description of Ziziphora), while others are too concise (e.g., description of the activity of Thymus vulgaris).
6) Line 194: „(…)with thymol, carvacrol, and linalool consistently identified as primary components - Add information about the concentration range of main constituents (the same comment for the other described OEs)
7) Table 1 should be cited in the text. The third and fourth columns could be combined as "Observed effect/Mechanism of action" because they sometimes present the same information. Add the information regarding from which part of the plants the observed effect was obtained.
8) Figure 3 should be placed after its first mention in the text.
9) Conclusion: „different EO” – should be EOs (also in line 586); „Depending on the EO content (…)” – should be probably „EOs composition”
Minor comments:
There are some editorial errors: unnecessary capital letters e.g. „Sylvestris”, lack of italics for the name of the plant (e.g. line 195, 206, 327 ….), ml should be „mL”
Author Response
Please see the answers to your comments attached and below.
This is a quite well-written manuscript devoted to the neuroprotective activity of different essential oils. The topic is interesting.
However, the manuscript needs some improvements as listed below:
The aim of the review article should be clearly stated in the Introduction.
Answer: A precise aim of the study was added.
The methodology used for preparing the review (lines 119-123) should be placed in the Introduction. "Papers outside the scope of this review, found by this search because certain keywords were mentioned in the text, were excluded." - the exclusion criteria should be specified in detail.
Answer: More precise exclusion criteria was added, but the main one, which was excluding more than 90% of publications, was described.
Figures should be placed as near to their first mention in the text as possible to facilitate easier following
Answer: As suggested, Figures were placed closer to the text where they are first mentioned.
The quality of Figure 1 should be improved. Fonts should be larger because they are hardly visible.
Answer: As suggested, the quality of Figure 1 was improved.
Section 4 should be divided into subsections for clearer organization (e.g., taking into account families). Moreover, in my opinion, there is too much unnecessary information in this section (see e.g., lines 182-193), and sometimes information directly linked with the aim of the study is too sparse (see e.g., lines 199-202). Some parts are rich in details (e.g., description of Ziziphora), while others are too concise (e.g., description of the activity of Thymus vulgaris).
Answer: The subsections, although not clearly highlighted have been previously established when constructing this part of the manuscript. Now we have clearly divided the manuscript according to plant families, usage of plants in Chinese medicine and as foodstuff. Also, unnecessary information stated by the reviewer are deleted and some parts are shortened and rewritten. Some families have more information given about them since they have been studied in more details and/or have higher ethnomedicinal background linking them with CNS diseases.
Line 194: „(…)with thymol, carvacrol, and linalool consistently identified as primary components - Add information about the concentration range of main constituents (the same comment for the other described OEs)
Answer: In the mentioned sentence the constituents were just highlighted, their concentration is not relevant in that particular case. In the next mention of the T. vulgaris, with the varieties depending on the constituents, the percentage was added.
Table 1 should be cited in the text. The third and fourth columns could be combined as "Observed effect/Mechanism of action" because they sometimes present the same information. Add the information regarding from which part of the plants the observed effect was obtained.
Answer: The mention of the Table 1 has been added to the text. We initially had the same idea of combining the two columns, however, the results and the conclusions of the studies do not overlap sometimes, and the authors in some cases only mention the obtained result, refraining from any conclusion on the mechanism since it is not possible based on their results. In other cases, however, authors examined a certain key molecule in the cases of events that are associated with inflammation/degeneration and thus they proved the exact mechanism. To our interpretation these two should be separate, since combining might potentially result in misleading information. We have added the information about the plant part used to extract essential oil where it was clearly stated in the text of the manuscript.
Figure 3 should be placed after its first mention in the text.
Answer: The figure has been moved.
Conclusion: „different EO” – should be EOs (also in line 586); „Depending on the EO content (…)” – should be probably „EOs composition”
Answer: Corrections were made as suggested.
Minor comments:
There are some editorial errors: unnecessary capital letters e.g. „Sylvestris”, lack of italics for the name of the plant (e.g. line 195, 206, 327 ….), ml should be „mL”
Answer: The suggested errors have been corrected.

Round 2
Reviewer 2 Report
Comments and Suggestions for Authors
1) At the same time, J. grandifolium EO decreased NO production and IL-1β and 359 TNF-α (using Western blot) and prevented ROS generation within the cell, Exposure of BV-2 cells to 372 LPS led to an upregulation in IL-6, IL-18, IL-1β, and TNF-α transcription and secretion, the Greek letters in the two paragraphs are not italicized. Please carefully check the entire text again.
2) Table 1 still has minor issues, areal parts should change to aerial parts, multiple items need to be modified. rhyzom should change to rhizome.
3) Flower EO of J. grandifolium containing ten different classes (sesquiterpenoids, monoterpenes, diterpenes, The spelling here should be consistent and changed to sesquiterpenoids, monoterpenoids, diterpenoids.
Author Response
Answering your questions:
1) At the same time, J. grandifolium EO decreased NO production and IL-1β and 359 TNF-α (using Western blot) and prevented ROS generation within the cell, Exposure of BV-2 cells to 372 LPS led to an upregulation in IL-6, IL-18, IL-1β, and TNF-α transcription and secretion, the Greek letters in the two paragraphs are not italicized. Please carefully check the entire text again.
The whole text has been scanned and all the greek letters are now in italics
2) Table 1 still has minor issues, areal parts should change to aerial parts, multiple items need to be modified. rhyzom should change to rhizome.
Table has been reviewed and mistakes corrected
3) Flower EO of J. grandifolium containing ten different classes (sesquiterpenoids, monoterpenes, diterpenes, The spelling here should be consistent and changed to sesquiterpenoids, monoterpenoids, diterpenoids.
Spelling has been fixed to monoterpenes, sesquiterpenes, etc.
Reviewer 4 Report
Comments and Suggestions for Authors
Authors corrected the manuscript or they provided necessary explanation when the suggestions have been omitted. Manuscript can be published; however, a few editorial correction should be done.
Line 207: Thymus vulgaris should be abbreviated as T. vulgaris
Line 182: “Sylvestris” – unnecessary capital letter
Table 1: „L” should not be italized; „oilresin” - it should be: “oil resin” (italic is also unnecessary); „in floresence” – it should be probably „inflorescence”; „rhyzom” – it should be „rhizome”
Author Response
Aswering your comments:
Authors corrected the manuscript or they provided necessary explanation when the suggestions have been omitted. Manuscript can be published; however, a few editorial correction should be done.
Line 207: Thymus vulgaris should be abbreviated as T. vulgaris
It has been corrected.
Line 182: “Sylvestris” – unnecessary capital letter
It has been corrected.
Table 1: „L” should not be italized; „oilresin” - it should be: “oil resin” (italic is also unnecessary); „in floresence” – it should be probably „inflorescence”; „rhyzom” – it should be „rhizome”
It has been corrected.